# Potential of Aqueous Humor as a Liquid Biopsy for Uveal Melanoma

**DOI:** 10.3390/ijms23116226

**Published:** 2022-06-02

**Authors:** Deborah H. Im, Chen-Ching Peng, Liya Xu, Mary E. Kim, Dejerianne Ostrow, Venkata Yellapantula, Moiz Bootwalla, Jaclyn A. Biegel, Xiaowu Gai, Rishvanth K. Prabakar, Peter Kuhn, James Hicks, Jesse L. Berry

**Affiliations:** 1The Vision Center at Children’s Hospital Los Angeles, Los Angeles, CA 90027, USA; deborahi@usc.edu (D.H.I.); ppeng@chla.usc.edu (C.-C.P.); lixu@chla.usc.edu (L.X.); maryekim@usc.edu (M.E.K.); 2USC Roski Eye Institute, Keck School of Medicine, University of Southern California, Los Angeles, CA 90033, USA; 3Center for Personalized Medicine, Department of Pathology and Laboratory Medicine, Children’s Hospital Los Angeles, Los Angeles, CA 90027, USA; dostrow@chla.usc.edu (D.O.); vyellapantula@chla.usc.edu (V.Y.); mbootwalla@chla.usc.edu (M.B.); jbiegel@chla.usc.edu (J.A.B.); xgai@chla.usc.edu (X.G.); 4Department of Pathology and Laboratory Medicine, Keck School of Medicine of USC, Los Angeles, CA 90033, USA; 5Department of Biological Sciences, Dornsife College of Letters, Arts, and Sciences, University of Southern California, Los Angeles, CA 90007, USA; kaliappa@usc.edu (R.K.P.); pkuhn@usc.edu (P.K.); jameshic@usc.edu (J.H.); 6Norris Comprehensive Cancer Center, Keck School of Medicine, University of Southern California, Los Angeles, CA 90033, USA; 7Department of Aerospace and Mechanical Engineering, Viterbi School of Engineering, University of Southern California, Los Angeles, CA 90007, USA; 8Department of Biomedical Engineering, Viterbi School of Engineering, University of Southern California, Los Angeles, CA 90007, USA; 9Department of Biochemistry and Molecular Medicine, Keck School of Medicine, University of Southern California, Los Angeles, CA 90033, USA; 10The Saban Research Institute, Children’s Hospital Los Angeles, Los Angeles, CA 90027, USA

**Keywords:** aqueous humor, circulating-tumor DNA, liquid biopsy, uveal melanoma

## Abstract

Tumor biopsy can identify prognostic biomarkers for metastatic uveal melanoma (UM), however aqueous humor (AH) liquid biopsy may serve as an adjunct. This study investigated whether the AH of UM eyes has sufficient circulating tumor DNA (ctDNA) to perform genetic analysis. This is a case series of 37 AH samples, taken before or after radiation, and one tumor wash sample, from 12 choroidal and 8 ciliary body (CB) melanoma eyes. AH was analyzed for nucleic acid concentrations. AH DNA and one tumor wash sample underwent shallow whole-genome sequencing followed by Illumina sequencing to detect somatic copy number alterations (SCNAs). Four post-radiation AH underwent targeted sequencing of *BAP1* and *GNAQ* genes. Post-radiation AH had significantly higher DNA and miRNA concentrations than paired pre-radiation samples. Highly recurrent UM SCNAs were identified in 0/11 post-radiation choroidal and 6/8 post-radiation CB AH. SCNAs were highly concordant in a CB post-radiation AH with its matched tumor (r = 0.978). *BAP1* or *GNAQ* variants were detected in 3/4 post-radiation AH samples. AH is a source of ctDNA in UM eyes, particularly in post-radiation CB eyes. For the first time, UM SCNAs and mutations were identified in AH-derived ctDNA. Suggesting that AH can serve as a liquid biopsy for UM.

## 1. Introduction

Uveal melanoma (UM) is the most common primary intraocular cancer in adults [1]. Tumors arise from the uveal tract and can affect the choroid, iris, and ciliary body, with the latter two lesions being anatomically closer to the aqueous humor (AH). Conservative treatment consists of radiation, with enucleation reserved for the most advanced cases. Even when the intraocular tumor is successfully treated, approximately half of all patients with UM will develop metastases [2]. Unfortunately, metastatic UM is usually fatal within one year of symptom onset, as it is poorly responsive to chemotherapy and/or targeted therapy.

It is widely known that identifying tumor biomarkers stratifies the risk of metastatic disease and may help improve the earlier detection of metastases [3]. While once taboo, intraocular tumor biopsy via fine-needle aspiration biopsy (FNAB) is now part of the routine clinical workup for UM for prognostication and not for diagnosis. Tumor-derived prognostic molecular markers can be categorized into (1) gene expression profiles (GEP) (2) somatic copy number alterations (SCNA), or (3) mutations in key genes for UM oncogenesis. Recent multi-omic work has used these three categories to divide UM patients into four subsets of low (Type A), intermediate (Type B), and high (Types C and D) metastatic potential [4]. Clinically, Onken et al. found that this translated into a four-year risk of metastasis of 3% for Types A and B and upwards of 80% for Types C and D [5]. Cytogenetic characteristics include highly recurrent UM SCNAs such as monosomy 3, losses of chromosome arms 1p, 6q, 8p, and 16q, and gains of 1q, 6p, and 8q [6], with monosomy 3, 8q gain, and 1p loss being the most prognostically unfavorable [7]. Although the American Joint Committee on Cancer (AJCC) staging of UM does not currently include cytogenetic information and is based only on clinical observations, it has been shown that the addition of these molecular subsets yields a significant improvement in prognostication compared to the AJCC stage alone [8,9,10,11,12]. While FNAB is considered a safe procedure, there remain small risks of retinal detachment, subretinal hemorrhage, and vitreous hemorrhage, sometimes requiring further surgical intervention to repair [13,14]. Furthermore, direct tumor biopsy is often not repeatable without returning to the operating room. Given these risks, liquid biopsy has emerged as a less invasive alternative that also offers the ability to track genomic changes longitudinally over time without the need to return to the operating room.

Most liquid biopsy research in UM has focused on the blood as a biofluid source of circulating tumor cells and/or circulating tumor DNA (ctDNA). However, the low tumor fraction found in the blood due to the blood-ocular barrier limits the detection of prognostic biomarkers; thus, blood liquid biopsies for UM may be better utilized to detect systemic disease [15,16,17,18]. AH liquid biopsies have been investigated as an eye-specific alternative. However, previous studies of the AH in UM patients have only reported on cytokines and soluble HLA [19,20,21,22,23]. To our knowledge, no genomic prognostic biomarkers have been found in UM AH. Given that we have not only demonstrated the presence but also established the clinical utility of diagnostic and prognostic biomarkers in the AH of retinoblastoma eyes [24,25]. we hypothesized that the AH of UM eyes may similarly harbor tumor-derived nucleic acids to serve as a surrogate tumor biopsy in UM. Thus, the purpose of this study is to determine if ctDNA is present in the AH of UM eyes. The presence and prognostic significance of ctDNA isolated from the AH of UM patients has yet to be evaluated. Furthermore, eye-specific biopsy allows for repeatable testing near the primary tumor and may allow for detection of local recurrence, new avenues for prognostication, and objective markers of tumoral regression post-therapy.

## 2. Results

### 2.1. Patient Clinical Characteristics and Demographics

Overall, 37 AH and one tumor wash sample from 20 UM patients (20 eyes) were evaluated. Patient demographics and clinical characteristics are summarized in Table 1. A total of twelve (60%) choroidal and eight (40%) ciliary body (CB) tumor patients were included. All choroidal tumors (100%) were AJCC stage I or IIA and did not have concomitant CB involvement. Of CB tumors, 5/8 (62.5%) were AJCC stage I or IIA, while the other three were more advanced. A significant number of CB tumors were diagnosed at a more advanced AJCC stage than choroidal tumors (*p* = 0.003; Table 1). Of note, one choroidal UM patient underwent primary enucleation due to the tumor surrounding the optic nerve head. Results from a clinically indicated tumor biopsy, including UM mutation, preferentially expressed antigen in melanoma (PRAME) status, and GEP class is included (Table 1).

### 2.2. Evaluation of AH Nucleic Acid Content before and after Brachytherapy Radiation

Analysis of AH samples revealed measurable levels of nucleic acids in the majority of eyes with choroidal and CB melanomas, with the exception of RNA which was only detectable in post-radiation AH from four CB tumors (Figure 1C and Appendix A). Paired pre- and post-radiation AH samples from nine choroidal and eight CB melanoma eyes were analyzed. In both choroidal and CB AH, post-radiation AH samples had significantly higher concentrations of DNA and miRNA (choroidal: ssDNA, *p* = 0.035 and miRNA, *p* = 0.016) (CB: dsDNA, *p* = 0.023 and miRNA, *p* = 0.008) (Figure 1A,B,D).

### 2.3. Circulating Tumor DNA in AH

To determine the presence of ctDNA in the AH, shallow WGS was performed to profile SCNAs in all 37 AH samples (Appendix A). All pre-radiation AH SCNA profiles were neutral. SCNAs were found only in AH collected after brachytherapy radiation, with significantly more positive SCNAs found in CB post-radiation AH samples (6/8, 75%) than in choroidal post-radiation AH samples (0/11, *p* = 0.001; Figure 2B). Highly recurrent UM SCNAs of monosomy 3, 6p gain, 6q loss, and 8q gain were identified in SCNA-positive post-radiation AH samples (Figure 2C). A tumor wash (UM_019) was performed in a single case to determine whether this would be a feasible mechanism to obtain tumor samples for research (instead of a repeat tumor biopsy). There was high concordance of SCNA alterations between the post-radiation AH sample and the tumor wash collected before radiation (Pearson’s r = 0.978; Figure 2D).

Due to limited DNA concentration, four SCNA-positive AH samples (UM_005, 007, 012, and 013) were further evaluated for the presence of tumor variants in the *BAP1* and *GNAQ* genes. We analyzed the presence of SNVs using a pan-cancer predisposition panel and identified UM mutations in *BAP1* and *GNAQ* in 3/4 (75%) post-radiation AH samples (UM_005, 007, 012, and 013, Table 2). These were concordant with clinical tumor SNV testing from Castle Biosciences in two patients (UM_007 and 013). A mutation was identified in the AH from patient UM_005; however, this patient did not have FNAB results available from Castle Biosciences to determine concordance (Table 2). Additional gene variants were investigated to determine the potential of detecting mutations from the AH cfDNA beyond the *BAP1* and *GNAQ* genes (Appendix A).

## 3. Discussion

We examined 37 AH samples from 20 UM patients to investigate whether the AH liquid biopsy can serve as a surrogate for tumor biopsy. We hypothesized that the AH may contain ctDNA and may serve as a liquid biopsy for UM; these hypotheses arose from our work on another ocular cancer, retinoblastoma, wherein we and others have demonstrated that the AH is an enriched source of ctDNA [26,27,28,29,30] In this pilot study, we first demonstrated that there were measurable concentrations of nucleic acids (dsDNA, ssDNA, RNA, and miRNA) in the small volumes of AH that can be extracted from UM patients during plaque brachytherapy or enucleation. This is the first-time nucleic acids were quantified and characterized in the AH of UM patients. When comparing pre- and post-radiation AH samples, there was a significantly higher concentration of evaluated nucleic acids in post-radiation AH samples, most notably in patients with CB tumors (Figure 1A–D). We hypothesize that the increased nucleic acids in post-radiation AH are tumor-derived due to necrosis and lysis of tumor cells after radiation. Further, we hypothesize that the higher mean concentration of nucleic acids in CB AH compared to choroidal AH is likely due to the proximity of the tumor to the AH in these more anteriorly located neoplasms. Importantly, 17/20 (85%) UM tumors included herein were AJCC stage I or IIA tumors, suggesting utility in AH liquid biopsy even for smaller, early-stage UM.

Next, we determined whether the AH DNA was tumor-derived. Copy number variation profiling confirmed the presence of highly recurrent UM SCNAs [31] in post-radiation CB AH samples including monosomy 3, 6p gain, 6q loss, and 8q gain (Figure 2A). No SCNAs were found in post-radiation choroidal AH samples or any pre-radiation AH samples. Altered SCNA profiles were found in 0/11 and 6/8 (75.0%) post-radiation AH samples of choroidal and CB melanomas respectively, with a significant difference (*p* = 0.001) (Figure 2B). This suggests that AH liquid biopsy may be more useful for tumors that form anteriorly in the eye in proximity to the anterior chamber, thus allowing an increased amount of ctDNA to diffuse into the AH after radiation-induced necrosis of tumor cells.

In one patient, we attempted a tumor wash sample wherein the needle that had performed the tumor biopsy was washed with basic saline solution after it had already been flushed for clinical GEP analysis for Castle Biosciences. We were able to effectively identify ctDNA for research purposes without impacting the Castle Biosciences analysis required for the clinical care of this patient. We demonstrated near-complete concordance in the presence and amplitude of SCNAs between the genomic profiles from the post-radiation AH and tumor wash samples (Pearson’s r = 0.978; Figure 2D).

Finally, four post-radiation AH samples that harbored SCNAs (thus, a higher fraction of ctDNA) were evaluated for canonical UM mutations in *BAP1* and *GNAQ* genes. Common UM mutations *GNAQ*/*11* are thought to be an early event in the development of UM [32], and other mutations such as *BAP1* portend a worse prognosis [32]. We identified tumor-derived de novo UM mutations in 3/4 (75%) post-radiation AH samples in *BAP1* or *GNAQ* (Table 2). These were concordant with clinical tumor SNV testing results from Castle Biosciences in two patients with available clinical testing of tumor samples (Table 2).

Both the presence of SCNAs and SNVs in post-radiation AH samples provides strong evidence that dsDNA isolated from the AH is tumor-derived. SCNAs can be visualized by shallow WGS with lower nucleic acid input requirements with a lower limit of detection at ~5% tumor fraction [24]. Targeted next-generation sequencing for mutation detection requires higher nucleic acid input, which is a potential explanation for why some SCNA-containing samples with a known tumor SNV mutation were not detected. While further optimization is needed, these results suggest that the AH may serve as a valid surrogate biopsy to identify not only SCNAs but also specific UM mutations. As AH paracentesis is repeatable and can be performed in clinic, this may also facilitate the evaluation of other eye-specific biomarkers and longitudinal evaluation of local disease.

While we identified quantifiable amounts of dsDNA, ssDNA, miRNA, and RNA in the AH, the levels were much higher after radiation. In contrast to retinoblastoma, it appears that UM, known to be less necrotic, does not shed into the AH at the same level and may require radiation or other interventions to cause cell death and shedding [25,26]. This may be somewhat confounded by tumor size, as the majority of choroidal tumors in this cohort were relatively small. The larger and more anteriorly located CB tumors had a higher concentration of nucleic acids in post-radiation AH samples, which facilitated the detection of SCNAs and SNVs. Research has shown benefits in identifying tumor-derived prognostic molecular markers and even that eye color may play a role in the interaction of these biomarkers [33]. Thus, an AH liquid biopsy may serve as an adjunct to FNAB, for example, a biopsy performed at plaque placement and a paracentesis performed at the time of plaque removal, so that the complex prognostic association of tumoral biomarkers including GEP, SCNA, and mutations in key UM-related genes can be better understood and utilized.

A limitation of this study is that only 20 UM patients were evaluated, with the majority of choroidal tumors being relatively small. Additionally, because there are some risks associated with tumor biopsy, tumor was not available for research only analysis. We relied instead on clinically available information from Castle Biosciences testing. This was available for most, but not all patients. Future studies should maximize the availability of tumors as well as include larger study populations and larger choroidal tumors. Most liquid biopsy platforms center on identifying ctDNA, which was the aim of our study as well. However, we are also investigating other nucleic acids as potential biomarkers in UM. MiRNA has been detected in plasma, enveloped in extracellular vesicles, and discovered here in the AH; miRNA has been investigated as a significant target in many cancers, and future studies should explore its role in UM [34].

## 4. Materials and Methods

This investigation was a case series study at a tertiary care hospital (University of Southern California Roski Eye Institute). Samples were taken between August 2020 and May 2021.

### 4.1. Patient and Specimen Characteristics

This study included a convenience sample of 20 UM patients at the University of Southern California Roski Eye Institute from whom written informed consent for an AH sample was obtained. All samples consisted of ~0.1 mL of AH extracted via clear cornea paracentesis at the end of surgery for brachytherapy plaque placement (pre-radiation), brachytherapy plaque removal (post-radiation) or after enucleation without radiation. We include 37 AH samples from 20 UM eyes: a total of 17 matched AH samples pre-radiation and post-radiation, two AH samples post-radiation only, and one AH sample after enucleation without radiation. Radiation methods in the treatment of UM have been detailed and published previously [35] Genomic testing results were coded and maintained separately from clinical data and thus did not alter patient treatment for all participants.

### 4.2. Specimen Collection and Storage

A clear corneal paracentesis with a 30-gauge needle was performed to extract ~0.1 mL of AH from UM eyes during clinically indicated surgery to treat UM. The extraction method has been described in detail and published previously by our group for specimen collection from retinoblastoma eyes [26]. Briefly, needles only entered the anterior chamber via the clear cornea at the limbus and did not make contact with the iris, lens, vitreous, or UM tumor. Samples were stored on dry ice immediately and transferred to −80 °C within hours of extraction. Routine FNAB with either a 25- or 27-gauge needle was conducted on 14 patients for mutational analysis and 15 patients for GEP and PRAME status which was performed at Castle Biosciences (Phoenix, AZ, USA). In one patient, the same tumor biopsy needle was washed with basic saline solution separately to obtain a tumor wash sample.

### 4.3. Analysis of Nucleic Acid Content in the AH

Nucleic acids (dsDNA, ssDNA, RNA, and miRNA) were assayed using Qubit Assay Kits (Thermo Fischer, Waltham, MA, USA), which measures the concentration of the assayed nucleic acid with the Qubit Fluorometer (Thermo Fisher, Waltham, MA, USA) following the manufacturer’s manual.

### 4.4. Genomic Analysis of Samples

All samples underwent DNA isolation, sequencing, and analysis within 1 month of collection, as consistent with established methods of SCNA analysis [1,2,3]. Briefly, cell-free DNA of AH was isolated with the QIAamp circulating nucleic acid kit (QIAGEN, Hilden, Germany), and DNA from FNAB was isolated with the QIAamp DNA blood mini kit (QIAGEN, Hilden, Germany). Isolated DNA was used to prepare whole-genome libraries with the QIAseq Ultralow Input Library Kit (QIAGEN, Hilden, Germany) followed by 2 × 150 bp paired end shallow 0.1–0.3x whole-genome sequencing (WGS) for copy-number alteration profiling. SCNAs were considered to be present at 20% deflection from a baseline human genome, which is based on liquid biopsy analyses that have been previously established [26,36].

### 4.5. Single Nucleotide Variants (SNV) Analysis of Samples

For four SCNA-positive AH samples (UM_005, 007, 012, and 013), the same sequencing libraries then underwent targeted resequencing for mutation detection using a customized hybridization panel laboratory developed test (Twist Bioscience, San Francisco, CA, USA) at the Children’s Hospital Los Angeles Center for Personalized Medicine that covers *BAP1* and *GNAQ* gene exon regions. Bioinformatics analysis was performed in parallel and blind to the clinically available tumor testing (Castle Biosciences, Phoenix, AZ, USA). Results were compared to tumor SNV results in all but one patient for whom AH analysis was performed but there was no matched clinical analysis as insurance refused to cover the Castle testing.

### 4.6. Statistical Analysis

Categorical variables were compared using the Fisher’s exact test or linear-by-linear association test as indicated. Continuous variables were summarized as the mean ± standard error and percentages and non-normally distributed variables were compared by the Mann–Whitney U test. Paired t-test was used for pre- and post- paired AH sample comparison. All statistical tests were two-tailed, and *p* < 0.05 was considered statistically significant. *p*-values are represented as: *, *p* < 0.05; **, *p* < 0.01; ***, *p* < 0.001. The Pearson’s coefficient was calculated by comparing the segmented ratio to medians at 5k bins. All statistical analyses and plots were conducted using the Prism 8 (GraphPad, San Diego, CA, USA).

## 5. Conclusions

In conclusion, this study demonstrates that the AH is a source of ctDNA in UM, with a significantly higher yield of nucleic acids after radiation. Given the significantly higher concentration of nucleic acids in CB AH compared to choroidal AH, AH biopsy may be more useful in anteriorly located tumors. To our knowledge, this is the first study determining that (1) tumor nucleic acids are present and quantifiable in the AH of UM eyes and that (2) UM SCNAs and mutations can be identified from the AH. These results suggest that the AH can serve as a liquid biopsy for UM, especially in CB tumors. With further investigations, this novel AH liquid biopsy platform may allow clinicians to better prognosticate, monitor disease progression, and investigate local intraocular biomarkers for UM.

## 6. Patents

Drs. Berry, Xu and Hicks have filed a patent application entitled: Aqueous humor cell free DNA for diagnostic and Prognostic evaluation of Ophthalmic Disease.

## Figures and Tables

**Figure 1 ijms-23-06226-f001:**
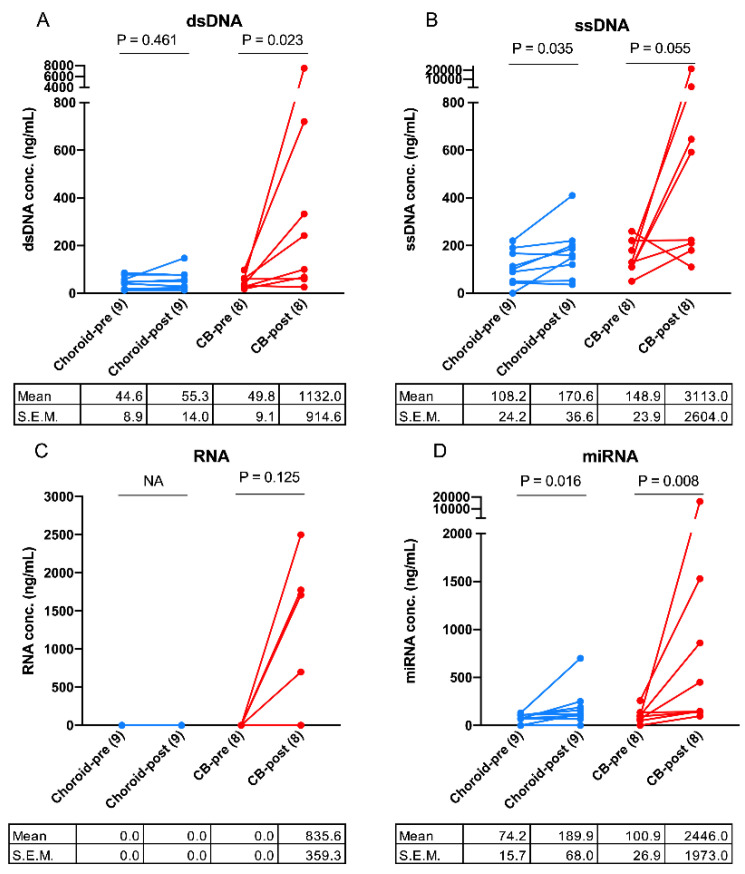
**Quantification of cell-free nucleic acids in UM aqueous humor samples before (pre-) and after (post-) radiation.** Concentration of (**A**), double-stranded DNA (dsDNA), (**B**), single-stranded DNA (ssDNA), (**C**), RNA and (**D**), microRNA (miRNA), were determined in 18 (9-paired) choroidal melanoma (choroidal) and 16 (8-paired) ciliary body (CB) melanoma AH samples. AH samples were grouped by collection time at pre- and post-radiation. P values were calculated by paired t-test. Mean and standard error of the mean (S.E.M.) were indicated. NA, not-available.

**Figure 2 ijms-23-06226-f002:**
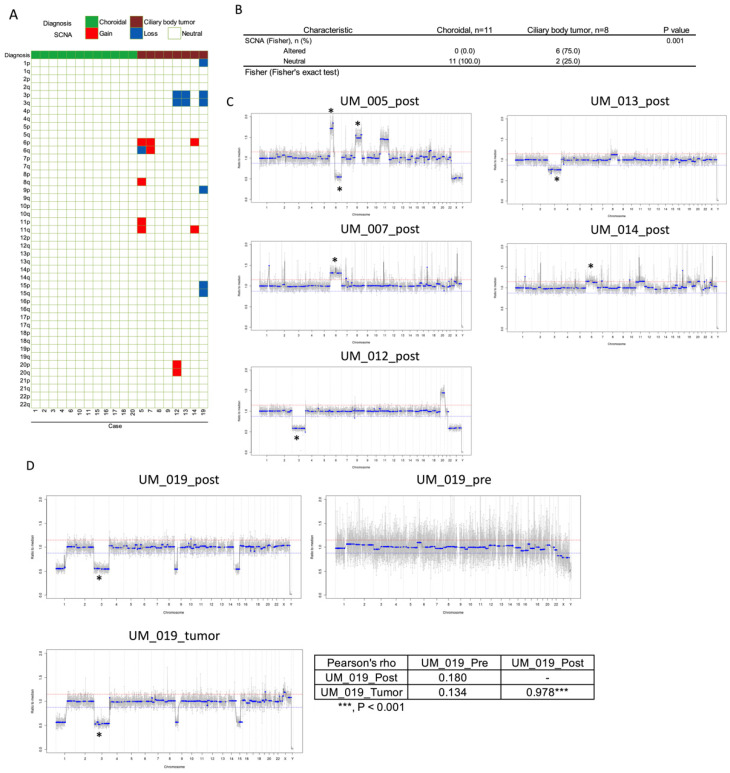
**Somatic copy number alterations in UM post-radiation aqueous humor samples.** (**A**) Schematic heatmap of the somatic copy number alterations (SCNA) in UM post-radiation AH samples. Gains and losses of chromosomes are shown. (**B**) Comparison of SCNA status (altered/neutral) between choroidal UM (*n* = 11) and ciliary body tumor (*n* = 8) in post-radiation samples. P value was calculated by the Fisher’s exact test. (**C**,**D**) Altered SCNA profiles identified from 6 UM post-radiation AH samples (UM_005, 007, 012, 013, 014 and 019). Highly recurrent UM SCNAs (monosomy 3, 6p gain, 6q loss and 8q gain) are indicated with an * on DNA profiles. (**D**) Consistency of the copy number variation (CNV) profile between UM_019_pre, UM_019_post and its corresponding tumor wash sample. Pearson’s correlation coefficient compared at 5k bins between each sample is indicated.

**Table 1 ijms-23-06226-t001:** Clinical characteristics of choroidal and ciliary body melanoma patients.

Characteristic		Choroidal, *n* = 12	Ciliary Body Tumor, *n* = 8	*p*-Value
Gender, *n* (%)				0.197
	Females	5 (41.7)	6 (75.0)	
	Males	7 (58.3)	2 (25.0)	
Eye, *n* (%)				0.650
	OD	5 (41.7)	5 (62.5)	
	OS	7 (58.3)	3 (37.5)	
Age at diagnosis, mean (± SD)		60.8 (12.5)	54.0 (15.5)	0.438
Eye Color, *n* (%)				0.999
	Light (blue, gray, green, hazel)	8 (66.7)	6 (75.0)	
	Dark (brown)	4 (33.3)	2 (25.0)	
Ciliary Body Involvement, *n* (%)				<0.001
	Yes	0 (0)	8 (100)	
	No	12 (100)	0 (0)	
AJCC Stage, *n* (%)				0.003
	I	9 (75.0)	1 (12.5)	
	IIA	3 (25.0)	4 (50.0)	
	IIB	0 (0)	2 (25.0)	
	IIIA, IIIB, IIIC	0 (0)	1 (12.5)	
	IV	0 (0)	0 (0)	
PRAME Status, known in 15 cases, *n* (%)				0.999
	Negative	7 (100)	7 (87.5)	
	Positive	0 (0)	1 (12.5)	
GEP Class, known in 15 cases, *n* (%)				0.876
	1A	5 (71.4)	6 (75.0)	
	1B	0 (0)	0 (0)	
	2	2 (28.6)	2 (25.0)	
Tumor Stage, *n* (%)				0.159
	T1	9 (75.0)	4 (50.0)	
	T2	3 (25.0)	3 (37.5)	
	T3	0 (0)	1 (12.5)	
	T4	0 (0)	0 (0)	

AJCC, American Joint Committee on Cancer; GEP, gene expression profile; PRAME, preferentially expressed antigen in melanoma; SD, standard deviation; Categorical variables (Gender, Eye, Eye color, Ciliary body involvement, PRAME, and GEP Class) were compared by Fisher’s exact test. Continuous variables (age at diagnosis) were compared by the Mann-Whitney U test. AJCC Stage and Tumor Stage were compared by Linear-by-Linear association.

**Table 2 ijms-23-06226-t002:** Single nucleotide variant analysis of BAP1 and GNAQ in four CB patients.

Sample	BAP1 (VAF%)	GNAQ (VAF%)
ciliary body tumor		
UM_005_Tumor	NA	NA
UM_005_AH	ND	c.626A > T (42.9)
UM_007_Tumor	ND	c.626A > T (23.2)
UM_007_AH	ND	c.626A > T (40.9)
UM_012_Tumor	ND	c.626A > T (53.0)
UM_012_AH	ND	ND
UM_013_Tumor	c.830_831del (68.0)	ND
UM_013_AH	c.830_831del (81.8)	ND

VAF, variant allele frequency; NA, not-available, tumor biopsy not conducted; ND, non-detectable Mutation detection assay in UM AH samples after radiation compared to clinical tumor single nucleotide variant (SNV) testing. UM mutations *BAP1* and *GNAQ* were detected in 3/4 (75%) post-radiation AH. These were concordant with clinical tumor tissue SNV testing from Castle Biosciences in two patients with available clinical testing of tumor tissue samples.

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
