# Peer review of "Potential of Aqueous Humor as a Liquid Biopsy for Uveal Melanoma"

_ijms, 2022, doi:10.3390/ijms23116226_

Round 1
Reviewer 1 Report
The authors show the usefulness of liquid biopsy for uveal melanoma.
The investigation is interesting, however, I have some concerns to be discussed.
- As you mentioned, the study cohort is so small.
- What is the difference between uveal melanoma and other malignancies in the case you use the liquid biopsy? The current method is suitable for other malignancies?
- I would say that it is not suitable for using samples after radiation therapy.
- It is interesting to investigate the correlation between liquid biopsy samples and histological samples.
- What are the advantages and disadvantages of comparing conventional biopsy? Please discuss citing the following article.
Limitations and usefulness of biopsy techniques for the diagnosis of metastatic bone and soft tissue tumors. Annals of medicine and surgery 2021 https://doi.org/10.1016/j.amsu.2021.102581
Reviewer 2 Report
Im and Peng et al try to verify AH from UM eyes whether harbor tumor-derived nucleic acids, and whether can serve as a liquid tumor biopsy choice in UM. Overall, the study is well performed, but the evidences are not sufficient to support the hypothesis. There are still some points the author would need to address.
Major points:
- It’s better to include the positive control (biopsy samples) or negative control (healthy or non-UM patients) to verify the sensitivity and accuracy of AH nucleic acid can be served as a liquid biopsy in UM?
- The authors try to reveal whether AH nucleic acid can be served as a liquid biopsy for UM. However, according to figure 1, there is not quite amount of detectable nucleic acid in choroid UM or pre-irradiation CB samples. Could the authors compare AH to circulating tumor DNA to show convincing statistic enriched in AH than circulating tumor DNA?
- BAP1 or GNAQ variants were not detectable in pre-radiation AH samples, does this mean AH can only consider as a liquid biopsy choice in post-radiation patients, which cannot be use as diagnostic choice.
Round 2
Reviewer 1 Report
The authors answered well, so the manuscript is suitable for publication.
Reviewer 2 Report
No more comments.